# PRANK: motion Prediction based on RANKing

**Yuriy Biktairov** [*][†]   **Maxim Stebelev**   **Irina Rudenko** [†]   **Oleh Shliazhko**   **Boris Yangel**
Yandex Self-Driving Group
{ybiktairov, mstebelev, irina-rud, olmer, hr0nix}@yandex-team.ru

## Abstract

Predicting the motion of agents such as pedestrians or human-driven vehicles is one of the most critical problems in the autonomous driving domain. The overall safety of driving and the comfort of a passenger directly depend on its successful solution. The motion prediction problem also remains one of the most challenging problems in autonomous driving engineering, mainly due to high variance of the possible agent's future behavior given a situation. The two phenomena responsible for the said variance are the multimodality caused by the uncertainty of the agent's intent (e.g., turn right or move forward) and uncertainty in the realization of a given intent (e.g., which lane to turn into). To be useful within a real-time autonomous driving pipeline, a motion prediction system must provide efficient ways to describe and quantify this uncertainty, such as computing posterior modes and their probabilities or estimating density at the point corresponding to a given trajectory. It also should not put substantial density on physically impossible trajectories, as they can confuse the system processing the predictions. In this paper, we introduce the PRANK method, which satisfies these requirements. PRANK takes rasterized bird-eye images of agent's surroundings as an input and extracts features of the scene with a convolutional neural network. It then produces the conditional distribution of agent's trajectories plausible in the given scene. The key contribution of PRANK is a way to represent that distribution using nearest-neighbor methods in latent trajectory space, which allows for efficient inference in real time. We evaluate PRANK on the in-house and Argoverse datasets, where it shows competitive results.

## 1   Introduction

This paper focuses on the problem of predicting the motion of agents surrounding a self-driving vehicle, such as pedestrians and vehicles driven by humans. As any vehicle, a self-driving vehicle needs certain time to change its speed, and sudden changes in speed and acceleration may feel very uncomfortable to its passengers. The motion planning module of a self-driving vehicle thus needs to have a good idea of where nearby agents might end up in a few seconds, so that it can plan for the vehicle to maintain safe distance. It can also greatly benefit from understanding when the situation is inherently multimodal, and multiple distinct futures are likely, as in such situations a self-driving vehicle might maintain extra caution until the situation becomes more clear.

One way to describe agent's future motion, which we choose to follow in this paper, is to represent it as a probability distribution over agent's future trajectories conditioned on the information about the scene up to the time the prediction was made. In order for this representation to be useful for the motion planning subsystem, it has to satisfy several requirements:

---

[*]Skolkovo Institute of Science and Technology

[†]Moscow Institute of Physics and Technology, yuriy.biktairov@phystech.edu

- The distribution needs to be computed with low latency, so that predictions can quickly react to new information such as a sudden change in agent's acceleration, and promptly inform the planning subsystem about the situation change;

- The distribution representation must provide efficient access to the expected and most likely future agent positions, so that the planning subsystem can use this information in real-time;

- There must be a convenient way to quantify future uncertainty and multimodality, if there's any. One way to achieve that is to explicitly enumerate the modes of the distribution together with their probabilities and statistics such as means or, possibly, higher-order moments.

The main contribution of this paper is to propose an approach to the problem of agent's future trajectory prediction, which we call PRANK (motion Prediction based on RANKing). The proposed approach satisfies the requirements listed above and is suitable for use in a production self-driving pipeline. Unlike generative approaches to this problem such as [1] or [2], the proposed method is closer to ranking techniques in its nature and heavily builds on the success of metric learning methods coupled with approximate nearest neighbor search in computer vision [3, 4] and NLP [5, 6] applications. The proposed method is based on the following ideas:

- Trajectory predictions are selected by scoring a fixed dictionary (or bank, as we call it) of trajectories prepared in advance, such as a representative subset of all agent trajectories ever observed in real life by the perception system of a self-driving fleet;

- The model is factorized in such a way that it can greatly benefit from preprocessing the trajectory bank offline. Only the encoding of the scene needs to be performed in real-time, and the rest of inference is handled either by directly using approximate nearest neighbor search methods on the trajectory bank, or by simple Monte-Carlo estimation on top of its results.

## 2 Related work

The problem of predicting the motion of vehicles and pedestrians has attracted a lot of attention from both the academic community and autonomous driving industry. There are two general approaches to describing and predicting future motion: predicting the intent of an agent and predicting the future motion trajectory directly.

Intent-based approaches cluster possible future actions of an agent into a discrete set of possible outcomes, such as staying in lane vs. changing lane for a vehicle, or crossing the street vs. continuing to walk on a sidewalk for a pedestrian, and then predict likely outcomes given a scene. One example is [7], which attempts to infer whether a vehicle will turn left, turn right or drive straight at an intersection. The discrete outcomes can be further augmented with information such as a goal location, e.g. what is the intended merge location of a vehicle in a merge maneuver [8].

Intent-based approaches provide useful information to the motion planning subsystem of a self-driving vehicle, but are fundamentally limited in expressive power as they don't specify exact trajectories that the agent might take, so some guesswork still has to be made at the planning level. This is why in practice such approaches tend to be combined or completely replaced by motion trajectory prediction methods.

Motion trajectory prediction approaches either aim to directly predict the most likely future trajectories of an agent [9, 10, 11, 12] or output a continuous distribution over such trajectories [1, 13, 14, 15] or even produce a time-factorized distribution over possible agent locations as a probabilistic grid [16]. While these methods can potentially capture the complex nature of possible future motion, it comes at a cost of sophisticated training and inference procedures [13, 14] or using a relatively simple generative model [1, 15], which is good at capturing frequently occurring patterns of motion such as forward movement and smooth turns, but may fail to behave well in complex scenarios. Our aim is to propose a method that can overcome this limitations.

One work that shares some similarities with ours is CoverNet [17], where predictions are not generated by the model from scratch, but rather selected from a predefined set with a classification head. However, the trajectory set in this work is generated by a simplistic motion model, the method does not allow for adding new trajectories to the set without retraining the model and is limited in the number of trajectories it can handle.

## 3 PRANK Approach

We formulate the task of agent motion prediction as a probability modeling problem in a discrete-time trajectory space $\mathbb{R}^{M \times 2} \equiv \mathbb{T}$, in which agent trajectories are represented by 2D positions of the center of agent's bounding box at times sampled at $M$ regular intervals. The proposed method takes a scene description $q$ as an input and outputs $P(t|q)$, the posterior distribution over possible trajectories of the agent given the scene. The scene description we use through the paper is a rasterized bird-eye view of the scene $q \in \mathbb{R}^{C \times H \times W} \equiv \mathbb{Q}$ centered around the agent, which encodes present and past agent positions, road graph structure, traffic light states etc. However, the proposed method can be used with other scene encoders such as VectorNet [10]. The details of the feature rasterization and trajectory encoding schemes are given in section 4.2.

The family of distributions we use to model the posterior is heavily inspired by the DSSM ranking approach [5]:

$$P(t|q) \sim h(t) \exp\left[ \alpha f(q)^T g(t) \right],\tag{1}$$

where $\alpha$ is a learnable parameter and $f : \mathbb{Q} \to \mathbb{S}_d$, $g : \mathbb{T} \to \mathbb{S}_d$ are neural network encoders mapping tensor representations of scene and trajectory into a common latent space $\mathbb{S}_d$ — unit sphere in $\mathbb{R}^d$. The term $h(t)$ is an auxiliary distribution with bounded support, which is needed to make DSSM-style distribution normalizable over a continuous domain, such as the trajectory space $\mathbb{T}$ we are dealing with. The additional implications of having the term $h(t)$ in the posterior distribution will become clear in section 3.2.

There are several reasons why we choose to model the posterior distribution this way:

- the bilinearity of the posterior distribution with respect to scene and trajectory representations in the common latent space allows for efficient MAP inference using approximate nearest neighbor search methods [6] if the trajectory set induced by $h(t)$ is fixed in advance. That in turn allows us to compute the mean and the mode of the posterior efficiently, as described in section 3.4;

- if the trajectory set is fixed in advance, it can be limited to contain only physically plausible agent trajectories, which greatly simplifies the modeling problem for the neural network and prevents making completely wrong predictions in the areas of the problem space where the prediction quality is not very good. The trajectory set can also be enriched with trajectories corresponding to complex maneuvers, thus ensuring that such maneuvers can be described by the model;

- as shown by the success of the DSSM-like models in the NLP community [18, 19, 6], the bilinearity assumption allows to model very complex non-linear interactions and, thus, should not be a limiting factor;

- there exists a consensus [20] that problems where the goal is to score a potential solution are in general computationally simpler than problems where an answer needs to be generated from scratch and, thus, should be more approachable by neural networks. So formulating the motion prediction problem as a scoring problem might provide a boost of prediction quality, which is supported by the evidence presented in section 4.4.1.

### 3.1 Training Process

We optimize $\alpha$ and the parameters of neural networks $f$ and $g$ in (1) by maximizing the likelihood of the training set $D = \{(q, t)_i\}$, which consists of ground truth agent trajectories paired with the representation of the scene around the agent. The log-likelihood takes the form

$$\log L(D) = \sum_{(q,t) \in D} \log \frac{h(t) \exp\left[ \alpha f(q)^T g(t) \right]}{Z(q, \alpha, f, g)},\tag{2}$$

where

$$Z(q, \alpha, f, g) = \int_{\mathbb{T}} h(t) \exp\left[ \alpha f(q)^T g(t) \right] dt$$

is the normalizing constant. As the exact value of $Z$ cannot be easily computed, we instead resort to a Monte-Carlo estimate

$$\hat{Z}_{MC}\left(q,\alpha,f,g\right) = \frac{1}{N}\sum_{t'\sim h}\exp\left[\alpha f\left(q\right)^T g\left(t'\right)\right], \tag{3}$$

which we compute by sampling a large number of trajectories from $h(t)$ for each training batch. The factor $h(t)$ in the numerator contributes a constant bias to the log-likelihood and thus can be ignored during training.

## 3.2 Choosing $h(t)$ and Trajectory Bank

In order to be a mode of the posterior distribution, a trajectory $t$ must satisfy the condition

$$0 = \frac{\partial}{\partial t}P\left(t|q\right) \sim \exp\left[\alpha f\left(q\right)^T g\left(t\right)\right]\frac{\partial}{\partial t}h\left(t\right) + h\left(t\right)\frac{\partial}{\partial t}\exp\left[\alpha f\left(q\right)^T g\left(t\right)\right].$$

MAP solutions found by the nearest neighbor methods such as those used in [6] will only turn the second term of this expression to zero. So in order for these methods to be able to provide inferences close to the true posterior modes, we would like to choose $h\left(t\right)$ to be as uniform over its support as possible, so that $\frac{\partial}{\partial t}h\left(t\right)$ is small and the first term is close to zero as well.

As shown in sections 3.1 and 3.4, our training and inference procedures only require sampling from $h\left(t\right)$, so it can be defined via a generative process. One way to define $h\left(t\right)$ so that it is close to uniform and does not put mass on physically implausible trajectories, which we propose in this paper, is as follows:

- Apply a clustering procedure to all agent trajectories in the training set.
- Sample from $h\left(t\right)$ by first sampling a cluster index, and then randomly choosing a trajectory from the training set that belongs to that cluster.

This way every sampled trajectory will be physically plausible, as it has been followed by some agent at least once. The clustering procedure will ensure that trajectories with high prior probability, such as stationary trajectories or uniform forward movement, will not be sampled quite as frequently, thus making the induced trajectory distribution much closer to uniform. We use mini-batch K-means [21, 22] algorithm with Euclidean distance in the trajectory space $\mathbb{T}$ for clustering as it is fast and seems to work well enough in practice, although other clustering procedures such as complete-linkage clustering may induce more uniform distributions. A more detailed study of the effect of clustering can be found in section 4.4.1 and supplementary materials.

## 3.3 Trajectory Noise Model

To account for the fact that the true trajectory of an agent may not be present in the trajectory set, as well as for the imperfection of the perception system that was used to obtain the trajectories in that set, we also propose to add noise to the PRANK model. To be precise, we propose to replace (1) by

$$P_{noise}\left(t|q\right) = \int_{\hat{t}\in\mathbb{T}}P\left(t|\hat{t}\right)P\left(\hat{t}|q\right)d\hat{t}, \tag{4}$$

where $P\left(\hat{t}|q\right)$ is given by (1), and

$$P\left(t|\hat{t}\right) \sim \exp\left(-\beta\|t-\hat{t}\|^s\right) \tag{5}$$

is the noise model. We use $s = 1$ throughout the paper. It is worth noting that the original model (1) can be obtained as a limit of (4) as $\beta \to \infty$. Maximizing the log-likelihood under the latent noise model requires yet another MC estimate

$$P_{noise}\left(t|q\right) \approx \frac{1}{N}\sum_{\hat{t}\sim h}P\left(t|\hat{t}\right)\frac{1}{\hat{Z}_{MC}}\exp\left[\alpha f\left(q\right)^T g\left(\hat{t}\right)\right], \tag{6}$$

which can be computed by reusing the samples used for estimating (3) during training.

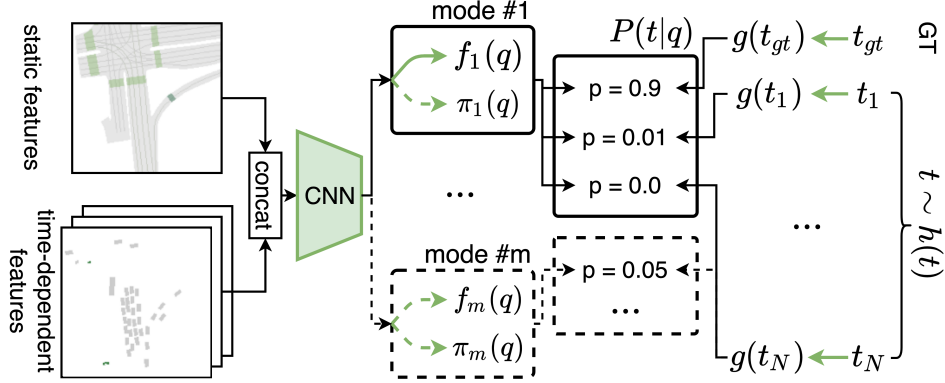

Figure 1: An illustration of the training process.

## 3.4 Inference Strategies

Here we explain how to perform inference efficiently in the proposed model (1) and its extension (4). Inference procedures described below require access to a representative sample of trajectories that have substantially non-zero probability under (1). To achieve that, we sample a large trajectory bank (up to 2M samples in our experiments) from $h(t)$, compute their embeddings $g(t)$ and use them to build an index optimized for approximate max inner product search. With such index we can quickly find almost all trajectories in the bank that have large value of (1) given a scene embedding $f(q)$, provided that the restrictions on $h(t)$ described in section 3.2 are met. We use Faiss [23] library to build and query the index.

**Estimating Mode** As explained in section 3.2, top-1 result from max inner product search is a reasonable approximation to the mode of (1), but only if $h(t)$ is close to uniform. As for the model with trajectory noise described in section 3.3, a procedure analogous to mean shift [24] can be used to maximize the estimate (6). In this case max inner product search provides samples corresponding to non-zero terms of this sum, provided that the number of results returned is large enough. This procedure correctly accounts for $h(t)$ not being completely uniform, giving the noisy version of the proposed model another justification.

**Estimating Mean** We can use the following estimate for the mean of the trajectory noise model:

$$
\mathop{\mathbb{E}}_{t \sim P_{noise}(t|q)}[t] = \int_{t \in \mathbb{T}} t \int_{\hat{t} \in \mathbb{T}} P\left(t|\hat{t}\right) P\left(\hat{t}|q\right) d\hat{t} dt \approx
$$

$$
\approx \int_{t \in \mathbb{T}} \frac{1}{ZN} \sum_{\hat{t} \sim h} t P\left(t|\hat{t}\right) \exp\left[\alpha f\left(q\right)^T g\left(\hat{t}\right)\right] dt = \frac{1}{ZN} \sum_{\hat{t} \sim h} \hat{t} \exp\left[\alpha f\left(q\right)^T g\left(\hat{t}\right)\right],
$$
(7)

provided that the noise distribution has its mean at $\hat{t}$. This allows us to use weighted average of the results of the max inner product search as an estimate of the posterior mean.

**Sampling from the posterior** of the trajectory noise model can be performed by following the generative process defined by (4): first picking a trajectory from the max inner product search results in proportion to $\exp\left[\alpha f(q)^T g(t)\right]$ and then sampling its noisy version using (5). However, the noise model (5) is rather non-restrictive and may produce physically implausible trajectories. So if realistic samples are needed, one should stick to samples from the model without noise.

## 3.5 Multimodal Generalization

When dealing with the prediction of future, it is not uncommon to be faced with a situation in which multiple distinct futures are likely. In order to handle such situations in the autonomous driving pipeline it is often desirable to explicitly reason about the number of modes, their probabilities and expectations. And while the models presented in the paper earlier are capable of capturing multimodal

posterior distributions, they don't provide direct access to this information. In order to address this limitation, we also propose a straightforward mixture model extension

$$P_m\left(t|q\right) = \sum_{k=1}^{m} \pi_k\left(q\right) \frac{1}{Z\left(q, \alpha_k, f_k, g\right)} h\left(t\right) \exp\left[\alpha_k f_k\left(q\right)^T g\left(t\right)\right],\tag{8}$$

where $\pi_k$ are mixture weights having $\sum_k \pi_k = 1$, and $f_k$ are mode-specific embedding generators. Both $\pi_k$ and $f_k$ can be implemented as separate heads on top of a common feature extractor network. The training procedure described in 3.1 can be straightforwardly extended to handle such models. As for the inference, the strategies discussed in the section 3.4 can be applied on a per-mode basis, with their results aggregated across modes if needed.

## 4  Experiments

In this section we describe the experimental setup, such as datasets, metrics, neural network architectures and training parameters. We then present an experimental evaluation of the proposed method, including an ablation study.

### 4.1  Datasets

While the proposed method can potentially be used to predict movement for other kinds of agents, for the sake of clarity and due to availability of public data we restrict ourselves to predicting the motion of vehicles. We present results on two datasets: the publicly available Argoverse dataset [25] and a much larger in-house dataset.

**In-house dataset** has been collected by recording rides of a fleet of self-driving vehicles over a course of 7 months. It contains about 1591K scenes for training, 11K for validation and 120K for test. Every vehicle that has been observed for at least 5 seconds and is not parked is considered a prediction target. A 5-second long ground truth trajectory sampled at 5Hz is available for such vehicles. That leaves us with about 6 prediction targets per scene on average. We don't perform any additional filtering or re-balancing of the dataset. Each scene also has 3 second long history data available. It includes positions, velocities and accelerations of all pedestrians and vehicles in the scene, as well as traffic light states; everything is as observed by the perception subsystem, sampled at 10Hz. HD map data in form of lanes, crosswalks and road boundaries, as well as ego vehicle state are also available.

**Argoverse** [25] is a public dataset designed for comparing vehicle trajectory prediction approaches. It consists of 333K scenes split into 211K scenes for training, 41K for validation and 80K for test. Each scene has a dedicated vehicle which is the prediction target. The task is to predict 3 second long trajectories given 2 second long history data, such as past agent locations and HD map. Both ground truth and history positions are sampled at 10Hz.

### 4.2  Neural Network Architecture and Training Details

We encode the scene as a set of $400 \times 400$ floating-point feature maps representing various aspects of the bird-eye view of the scene with the spatial resolution of 0.5 meters. All feature maps share the same reference frame, which is chosen such that the agent of interest is located in the center of the map at the prediction time and the Y axis of the map is aligned with agent's orientation. Each map represents some aspect of the scene such as lane or crosswalk locations, agent presence, speed or acceleration at a particular timestamp, lane availability induced by traffic light states etc. Feature maps are concatenated along the channel dimension to form the input of the scene encoder $f$. We use 7 timestamps for time-dependent feature maps with 2Hz rate for the in-house dataset, which, combined with time-independent maps gives us a total of 71 input channels. For the Argoverse dataset we use 10 timestamps at 5Hz rate and 62 input channels respectively.

A trajectory is represented as a $\mathbb{R}^{M \times 2}$ vector of agent's $(x, y)$ locations sampled at $M$ regular time intervals. $M$ is 25 for the in-house dataset and 30 for the Argoverse dataset. All trajectories are expressed in the reference frame where agent's position at prediction time is $(0, 0)$ and the agent is oriented along the first dimension.

| Table 1: Ablation experiments | | | |
| --- | --- | --- | --- |
| Model | ADE | FDE | LL |
| base | **1.302** | **2.968** | -2.12 |
| no noise | 1.302 | 3.017 | -2.10 |
| no history | 1.339 | 3.056 | -2.24 |
| no rebalancing | 1.434 | 3.330 | -2.62 |
| 2 modes | 1.310 | 2.990 | -1.70 |
| 3 modes | 1.308 | 2.985 | -1.61 |
| 5 modes | 1.309 | 2.986 | **-1.56** |

| Table 2: Inference approaches comparison | | | | |
| --- | --- | --- | --- | --- |
| Model | Inference | ADE | FDE | Hit rate (0.5m) |
| no noise | mode, top-1 | 1.450 | 3.318 | 0.176 |
| | mean, top-150 | 1.302 | 3.017 | 0.191 |
| | mean, top-500 | 1.291 | 2.992 | 0.190 |
| | mean, top-1000 | **1.287** | 2.983 | 0.189 |
| noise | mean shift, top-150 | 1.314 | 2.995 | **0.193** |
| | mean shift, top-500 | 1.311 | 2.987 | **0.193** |
| | mean shift, top-1000 | 1.309 | 2.984 | **0.193** |
| | mean, top-150 | 1.302 | 2.968 | 0.192 |
| | mean, top-500 | 1.297 | 2.956 | 0.192 |
| | mean, top-1000 | 1.295 | **2.952** | 0.191 |

We use a shallower and narrower version of Inception-ResNet-v1 [26] with a total of 716K parameters as the scene encoder $f$. The trajectory encoder $g$ is a sequence of fully-connected layers with skip connections and batch norm layers, having a total of 12 fully-connected layers and 36K parameters. We've experimented with other encoder architectures but found no significant improvement. The dimensionality of the shared trajectory-scene latent space is 64. Trajectory index size is 2M for the in-house dataset and 250K for Argoverse.

We use the Adam optimizer [27] with the batch size of 128 split between 4 GeForce 1080 GTX GPUs. The learning rate starts at $5e-4$ and is reduced by half every time there is no improvement in validation loss for 5 pseudo-epochs of 2000 batches each (1000 for Argoverse). Training our models on the in-house dataset until convergence in this mode takes about 5 days, training on the Argoverse takes 3 days.

## 4.3 Metrics

We report metrics commonly used in the trajectory prediction literature: mean displacement error on the last timestamp (*FDE*), mean displacement error over all timestamps (*ADE*), the fraction of agents for which maximum displacement error over all timestamps is less than 0.5 meters (*Hit rate*). To assess the ability of various models to correctly describe multiple modes, we also introduce the *LL* metric, which evaluates a weighted set of predictions provided by a model by computing log-likelihood of the ground truth trajectory $t^{gt}$ under a simple probabilistic model:

$$ LL \left( \left\{ t_i^{pred} \right\}, \left\{ w_i^{pred} \right\} \right) = \log \left[ \sum_i w_i^{pred} \mathcal{N} \left( t^{gt} | t_i^{pred}, \sigma^2 I \right) \right]. $$

Here $w_i^{pred}$ must sum to 1 and $\sigma$ is set to 1. We believe this metric to be a much better indicator of multimodal performance than commonly used *DE@N* metrics that measure the error of the best prediction among the top $N$. The latter don't take prediction weights into account and can be easily manipulated by emitting a diverse set of predictions even if the model believes there being just one mode.

## 4.4 Experimental evaluation

### 4.4.1 In-house dataset

The performance of various versions of the proposed method on the in-house dataset can be seen in Table 1. Here we use the posterior mean estimated using top 150 max inner product search results as a prediction. It can be seen that disabling trajectory noise in the model (*no noise*) affects the performance. Using only one timestamp for time-dependent feature maps (*no history*) also affects the prediction quality, but not as much as one might think. Presumably, it's because a lot can be inferred about agent's intention from its acceleration even on a single timestamp. The hardest impact on the performance of the method comes from the lack of clustering-based rebalancing in $h(t)$ (*no rebalancing*), having it just emit trajectories from the training set uniformly, which aligns with our reasoning in section 3.2.

We also evaluate the performance of the multimodal version of the PRANK model with different number of modes. We use $\pi_k(q)$ as $w_k^{pred}$ and per-mode means calculated using (7) as $t_k^{pred}$ to

compute the LL metric, and $t_1^{pred}$ for ADE and FDE. It can be seen that while these models incur almost no hit in performance in terms of displacement, the mutimodality metric monotonically improves with the number of modes.

We evaluate the effect that the inference procedure has on the prediction quality and show the results in Table 2. As expected, picking the mode of the noisy model as a prediction performs better in terms of hit rate, which the mode directly optimizes. Poor hit rate of the top-1 search result for the model without noise indicates that $h(t)$ induced by clustering still isn't uniform, and, thus, has to be explicitly accounted for. Using the mean gives a better ADE and FDE metrics for both models. As the quality of Monte-Carlo estimates grows with the number of max inner product search results used, so do the metrics, although the effect quickly saturates. Hit rate degrades with the top size when using mean, as predictions get averaged between different modes more and more, and, thus, move further away from any of the modes. A more detailed study of how various top sizes affect the prediction quality can be found in the supplementary materials.

We also compare the proposed method with 2 baselines: a rule-based system that has been deployed in a real self-driving pipeline, and a generative decoder, which predicts future agent locations timestamp-by-timestamp using a 3-layer LSTM [32] with 64 hidden units. We argue that the latter is a good baseline for a new trajectory decoding scheme such as the one proposed in this paper, as state-of-the-art methods either use a simple fully-connected decoder [29, 31] or a RNN, which seems to work better [33]. The decoder takes the scene encoder output $f(q)$ as an input. We have evaluated 2 versions of the decoder: one predicts absolute agent positions in the reference frame at each timestamp, the other — relative movement from timestamp to timestamp. The decoders have been trained to minimize MSE loss between the prediction and the ground truth. As can be seen in Table 3, the proposed method has a clear advantage over models with the decoder, despite using the same information about a scene. That supports our argument that solution scoring can be preferable over generating a solution from scratch, as in the former case an easier problem needs to be solved. The rule-based system, which hasn't been using any automated tuning, does not perform well on the dataset.

We show the effect that the index size has on the prediction quality in Table 4, using posterior mean computed from 150 max inner product search results as a prediction. As expected, larger trajectory indices generally perform better in terms of prediction quality. Interestingly, there seems to be a sweet spot at about 250k trajectories. We hypothesize that the reason for its existence is that the index size affects inference in two ways:

- Larger indices have more appropriate trajectory candidates for a scene on average, so the quality should improve as an index grows;

- A larger index has more trajectories similar to a given trajectory, so max inner product search results become less diverse for a fixed top size, and larger top sizes might be needed to compensate.

In practical applications search results count and index size should be picked jointly to maximize the prediction quality within the inference time and memory constraints.

### 4.4.2 Argoverse dataset

We present the results of our method on the Argoverse dataset in Table 3.

Our method is among the top-3 methods (out of more than 80 participants) submitted to the Argoverse dataset challenge [34] in terms of *ADE@1* and *FDE@1* metrics[3], despite having no specific tuning for the competition. A direct comparison of the proposed method with the methods ahead is hard, as we use a more sophisticated trajectory decoding scheme, while they propose better scene encoders. One entry, which, just as our method, uses a rasterized representation of the scene is «uulm-mrm» [30, 31], which we surpass in quality. We also outperform the recently published VectorNet [10] approach, despite using a much less sophisticated representation of the scene.

We've also tried applying PRANK on the Argoverse with the trajectory set obtained from the in-house dataset. It, however, did not lead to an improvement despite the in-house dataset being much larger

| Table 3: Comparison with other methods | | | | | Table 4: Index size study | |
|---|---|---|---|---|---|---|
| Dataset | Model | ADE | FDE | | Index size | ADE |
| In-house | PRANK | **1.302** | **2.968** | | 2M | 1.302 |
| | LSTM (abs) | 1.428 | 3.152 | | 1M | 1.303 ±0.001 |
| | LSTM (rel) | 1.454 | 3.293 | | 500k | **1.298** |
| | rule-based | 2.594 | 5.992 | | 250k | **1.298** ±0.001 |
| | | | | | 100k | 1.300 ±0.001 |
| Argoverse | Jean [28] | **1.68** | **3.73** | | 50k | 1.306 ±0.001 |
| Leaderboard | _anonymous (LGN) [29] | 1.71 | 3.78 | | 20k | 1.325 ±0.001 |
| ADE@1 top | PRANK | 1.73 | 3.82 | | 10k | 1.354 ±0.005 |
| | poly | 1.77 | 3.95 | | 5k | 1.41 ±0.03 |
| | UAR | 1.86 | 4.09 | | 2k | 1.52 ±0.05 |
| Argoverse | VectorNet [10] | 1.81 | 4.01 | | 1k | 2.3 ±0.7 |
| Other results | PRANK (in-house trajectories) | 1.84 | 4.05 | | 500 | 2.7 ±1.3 |
| | uulm-mrm [30, 31] | 1.90 | 4.19 | | 150 | 5.3 ±1.4 |

and more diverse. We hypothesize it being due to a distribution mismatch, as the datasets have been collected in different countries, having somewhat different road layouts and agent's behavior.

## 4.5 Runtime performance

The proposed method takes about 200ms to produce predictions for 5 agents when running on GeForce RTX 2080 Ti and a single core of a modern CPU. Most of that time is consumed by rendering 5 sets of feature maps for the agents and running the scene encoder, while max inner product search and inference postprocessing takes less than 3ms for 150 search results. We've also experimented with sharing feature maps and scene encoding for all agents, as proposed in [1], which allowed us to reduce scene encoding costs and make predictions for 50 agents in the same amount of time with almost no impact on prediction quality.

## 5 Conclusion and Future Work

We have proposed a novel motion trajectory prediction method that is suitable for use in a real-time autonomous driving pipeline. The proposed method allows to quickly estimate the expected and most likely trajectories of an agent, as well as discover the number of plausible motion modes and their probabilities. On top of that, the proposed method avoids predicting trajectories that violate physical constraints. The key feature of the method is that it turns the problem of trajectory distribution modeling into the problem of ranking trajectories from a pre-built trajectory bank using approximate nearest neighbor search methods. All inference operations besides scene encoding can be then implemented as simple post-processing of the neighbor search results.

Possible directions of future work include

- exploring ways to train on agents that haven't been observed on every timestamp to overcome biases introduced by filtering out such agents;
- introducing dependency on scene description $q$ into $h$ so that we can additionally filter out trajectories that seem impossible in a particular situation, not just in general. One way this might be achieved is by exploiting the fact that every trajectory in the bank can be associated with a scene where this trajectory has been implemented, and a similarity measure between this scene and $q$ can be measured;
- exploring the effect of more sophisticated network architectures and alternative feature extraction pipelines such as [10] or [29] on the overall quality of predictions.

## 6 Broader Impact

Motion prediction methods can advance the development of the self-driving technology and, thus, inherit the impact associated with it. For a detailed overview of long-term effects of autonomous vehicles on the society we refer the reader to [35].

It should be explicitly mentioned that motion prediction methods can potentially advance the development of malicious self-driving agents that aim to create dangerous traffic situations or collide with a particular vehicle. The better such agents will be able to predict how other agents will behave, the harder it will be for them to avoid a collision. A joint effort of regulatory bodies, engineering and information security specialists is required to prevent such vehicles from ever appearing on public roads.

## Acknowledgments and Disclosure of Funding

This work has been funded solely by Yandex Self-Driving Group.

We would like to thank our colleagues from Yandex for fruitful discussions that led to the emergence of the ideas presented in this work.

## Footnotes

[3]The results have been obtained from the leaderboard on June 16th, 2020.

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
