[Supplementary Material]

# Supplementary materials

## 1   Number of samples needed to estimate $Z(q, \alpha, f, g)$

In order to find out how many samples are needed to get a good estimate of $Z(q, \alpha, f, g)$, we train a few models with varying number of samples and evaluate their performance on the in-house dataset. We use the noisy unimodal version of the model and use posterior mean estimated using 150 max inner product search results as a prediction.

The results are shown in table 1. It can be seen that $2^{14}$ samples are enough to get a good prediction in terms of ADE and LL metrics, but the error on the last timestamp continues to improve even when the number of samples is large.

Table 1: Number of samples needed to estimate $Z(q, \alpha, f, g)$

| | Metrics | | | | |
| --- | --- | --- | --- | --- | --- |
| # samples | ADE | ADE 90pct | FDE | FDE 90pct | LL |
| $2^{15}$ | 1.312 | **2.887** | **2.988** | **6.990** | -2.161 |
| $2^{14}$ | 1.310 | 2.900 | 2.991 | 7.056 | **-2.115** |
| $2^{13}$ | **1.309** | 2.898 | 2.996 | 7.025 | -2.130 |
| $2^{12}$ | 1.318 | 2.916 | 3.022 | 7.114 | -2.164 |
| $2^{11}$ | 1.319 | 2.917 | 3.026 | 7.138 | -2.166 |
| $2^{10}$ | 1.335 | 2.954 | 3.066 | 7.210 | -2.193 |

## 2   Number of search results needed to estimate the posterior mean

We also evaluate how the quality of the prediction based on the posterior mean depends on the number of max inner product search results used to estimate the mean. The results for the noisy unimodal model are shown in table 2. It can be seen that the prediction quality improves as the number of search results grows, although the gains become marginal after using about 2000 search results.

Table 2: Number of search results for posterior mean estimation

| | Metrics | | | | |
| --- | --- | --- | --- | --- | --- |
| # results | ADE | ADE 90pct | FDE | FDE 90pct | LL |
| 1 | 1.358 | 2.977 | 3.086 | 7.205 | -2.328 |
| 5 | 1.327 | 2.921 | 3.023 | 7.085 | -2.230 |
| 20 | 1.314 | 2.896 | 2.995 | 7.029 | -2.176 |
| 50 | 1.308 | 2.876 | 2.981 | 7.003 | -2.148 |
| 100 | 1.304 | 2.870 | 2.973 | 6.982 | -2.128 |
| 150 | 1.302 | 2.866 | 2.968 | 6.972 | -2.116 |
| 250 | 1.305 | 2.863 | 2.972 | 6.967 | -2.210 |
| 500 | 1.297 | 2.859 | 2.957 | 6.947 | -2.084 |
| 1000 | 1.295 | 2.857 | 2.952 | 6.94 | -2.065 |
| 2000 | **1.294** | **2.850** | 2.950 | 6.935 | -2.048 |
| 3000 | **1.294** | 2.851 | **2.949** | **6.931** | -2.039 |
| 5000 | **1.294** | 2.852 | 2.950 | 6.934 | **-2.029** |

## 3   The effects of clustering

We also try different clustering procedures and numbers of clusters for specifying $h(t)$ and find no significant effect on the results, as long as some clustering procedure is used. The results can be seen in Table 3. We argue that the specifics of the clustering procedure do not have a large effect on quality because the actual values of $h(t)$ become less relevant if we consider large enough number of max inner product search results, as we'll get most of the significant terms of the posterior anyway.

Table 3: A study of the effects of clustering

| Clustering procedure | N clusters | ADE | FDE |
|---|---|---|---|
| hierarchical compl.-linkage | 1k | 1.322 | 3.028 |
| | 5k | 1.314 | **3.007** |
| | 10k | 1.315 | 3.030 |
| k-means, implementation 1 | 10k | **1.313** | 3.011 |
| k-means, implementation 2 | 10k | 1.321 | 3.022 |
| k-means, implementation 3 | 10k | 1.322 | 3.031 |

## 4 Predictions examples

Some examples of our model's predictions are shown in figures from 1 to 5. For the sake of comparison, we also demonstrate the predictions of a generative model with an LSTM decoder. In all images ground truth future trajectories are green, predictions of our model are drawn in red and predictions of the generative model are blue. The ego vehicle is shown in gray. Cyan dots represent pedestrians. Red and green circles show the state of the corresponding traffic lights.

Figure 1: Examples of bad predictions made by our model.

Figure 2: Examples of our model predicting a better realization of a maneuver than the generative model.

Figure 3: Examples of our model picking a wrong mode.

Figure 4: Examples of our model picking the correct mode while the generative model is mistaken.

Figure 5: Examples of predictions of the bimodal PRANK model.