[Reviews · NeurIPS 2020]

Review 1

Summary and Contributions: The proposed work first samples a set of trajectories in the distribution of the training dataset. Then, a certain number of closer trajectories are found through max inner product search. The trajectories with a larger value is found.

Strengths: - The paper is well written and technically sound. - The way to find a set of trajectory candidates seems new. - The paper is well motivated (although the scope is limited).

Weaknesses: - In L36-43, the authors introduce three requirements: latency, accuracy for most likely trajectory, way to quantify future uncertainty and multimodality. However, the present work provides limited information on the latency and does not show any evaluation to address the way to quantify uncertainty. Although in Sec.4.5 the authors provide a runtime performance, it is hard to measure if the presented work shows low latency without providing any reference speed of existing methods or practical requirement in AD systems. Also, how do you quantify uncertainty? If such items are not properly addressed, the authors should rewrite L36-43. - By tweaking the sampling procedure, this work tries to make the trajectory distribution to be closer to uniform. For this, the clustering procedure is applied to all agent trajectories in the training set. Then, randomly sample trajectories from the randomly chosen cluster. This process assumes the following two factors: (1) the trajectories in the training dataset contain all possible trajectories in the test datset. The noise model (Sec.3.3) is somehow presented to address this issue, and the authors a study on this. In Table 1, between 'base' and 'no noise', there is no difference on ADE, less than 5cm improvement on FDE, and almost identical LL, which makes me still doubt about the impact of such noise model if it can actually cover the full test distribution. Any other evaluations? Also, how do you ensure that the noise model generates plausible trajectories? (2) The clustering method makes groups based on the trajectory or motion patterns. There is no study on the parameter K for clustering, which is critical for h(t). Without providing such factors, h(t) will not be closer to uniform, and therefore the proposed posterior distribution might not be properly modeled. - The motivation is limiting the scope of this work to be autonomous driving. However, I believe that the proposed method is generally usable and can be more highlighted when the scope is broaden to pedestrian trajectory prediction. Compared to vehicle trajectories, human motion is more diverse, uncertain, and multi-modal, which are addressed in this work. There exist widely used public benchmark datasets such as SDD, ETH, and UCY. Without such evaluation, I cannot see the generalizability of the proposed method since it is only evaluated using only one public dataset. - Although [A] was not available at the time this manuscript is submitted, I find it is very close to the proposed method. It will be interesting if the authors can review it and the advantage of the proposed method in Sec.2. - In Sec.4.3, what is the difference between the LL metric proposed in this work and LL metric used in [25]? - The presentation of evaluation results on Argoverse is not thorough. The authors should provide an analysis in depth using the metrics of the Argoverse leaderboard or by providing DE@t similar to [17]. - The results in Argoverse seems not very promising. [A] MANTRA: Memory Augmented Networks for Multiple Trajectory Prediction, CVPR 2020.

Correctness: The methodology seems correct. However, there are some claims that need to be reconsidered as mentioned as weaknesses.

Clarity: The paper is well written and technically sound.

Relation to Prior Work: Although [4] is reviewed in Sec.2, I think it is also close to this work in terms of that they find a set of anchors in the beginning. It will be better to review separately.

Reproducibility: Yes

Additional Feedback: I find several major concerns from this work, which can be better addressed: claims with no proper validation, lack of justification, shallow analysis, limited performance, lack of generalizability, etc. ==== After reading rebuttal ==== I still have a concern about this paper from the following two aspects: 1. Whether the current evaluations are sufficient for publication. 2. Whether the nearest neighbor search is good enough. In the former, the authors use one benchmark dataset to compare the proposed method with other approaches. However, considering that the proposed method does not require any specific labels, it is not convincing that the current evaluation is sufficient for publication. In the rebuttal, the authors responded that the small size of pedestrian datasets cannot build a trajectory bank. Then, the follow-up evaluation should be how the performance varies w.r.t. the size of training data to demonstrate their response. However, I cannot see such validation on the size. Without additional evaluation or study, the proposed method seems not generalizable or applicable to other datasets. In the latter, the authors didn't respond to my question about the noise model. If there is no impact on the noise addition, I believe that the proposed method is simply being a nearest neighbor retrieval framework with less contributions since the model only covers the observed trajectory distribution. In Table 1, between 'base' and 'no noise', there is no difference on ADE, less than 5cm improvement on FDE, and almost identical LL, which makes me still doubt about the impact of such noise model if it actually helps to cover the full test distribution. If so, I think the authors' underlying assumption cannot be addressed - the training dataset can cover the full test distribution by adding noise to the PRANK model, where this paper is built on. Although the proposed method sounds technical and mathematically well-formulated, these two factors should be solved to publish this work.


Review 2

Summary and Contributions: The paper proposed an algorithm to produce trajectories likely to be adopted in real life scenarios. PRANK takes rasterized bird-eye images of agent’s surroundings as an input and extracts features of the scene with a convolutional neural network. It then produces the conditional distribution of agent’s trajectories plausible in the given scene. The key contribution of PRANK is a way to represent that distribution using nearest-neighbor methods in latent trajectory space, which allows for efficient inference in real time.

Strengths: The proposed algorithm inherently predicts physically plausible trajectories as it considers clustered trajectories from real training data as candidates. Besides, the clustering of possible trajectories can also accelerate the testing phase.

Weaknesses: 1. The choice of trajectories used for clustering and the clustering itself might be an issue under certain conditions. Firstly, it might be hard to pick out representative pieces of trajectories for all kinds of scenarios. The future trajectory we want to predict might start from any moment of a full route, thus challenging the generalizability of selected ones. Secondly, the number of clusters might be hard to determine. As the pieces of trajectory to be predicted vary greatly, the cluster of future trajectories might be countless. Thirdly, even with sufficiently generalizable pieces of trajectories and a 'correct' cluster number, the clustering algorithm using Euclidean distance might still be questionable. For example, two trajectories might be close enough in that sense but one ends up turning left and the other ends up turning right. How to deal with such situations? It would be better to provide some insights on the above issues. 2. The comparisons with state-of-the-art approaches are not sufficient for a NeurIPS paper, especially for the in-house dataset. The existing baselines in the current paper are too simple. It would be better to add more SOTA baselines to demonstrate the advantages of the proposed approach.

Correctness: No major mistake was found.

Clarity: The paper is well written and easy to follow.

Relation to Prior Work: The authors provided a brief overview of related research, and pointed out the similarity and distinction between the proposed method and existing works.

Reproducibility: Yes

Additional Feedback: It is addressed that the proposed method can guarantee the feasibility of trajectories by selecting from trajectory candidates. What are the pros and cons of this method compared with those which directly use the kinematic model to confine the future trajectories? [1] https://arxiv.org/abs/1908.00219 [2] https://openaccess.thecvf.com/content_ICCV_2019/html/Bi_Joint_Prediction_for_Kinematic_Trajectories_in_Vehicle-Pedestrian-Mixed_Scenes_ICCV_2019_paper.html [3] https://ieeexplore.ieee.org/abstract/document/8813783 After reading the rebuttal, I would like to keep my original score since some of the concerns were not addressed sufficiently or missing.


Review 3

Summary and Contributions: The paper describes a novel method for predicting the trajectory of actors based on historical scene information.

Strengths: The novelty of the contribution is the formulation of the problem as a nearest neighbour search from a fixed length dictionary of trajectories. This has the benefits of efficiency, plausibility of trajectories, and potentially better solutions.

Weaknesses: The paper would have been even stronger, with a substantial improvement on Argoverse benchmark. The authors hint that with specific tuning they would have achieved better results but that is just speculations.

Correctness: Claims, methods and methodology is correct.

Clarity: The problem is well explained, the logical structure is adequate. It's a well written and very clear paper. Congrats to the author, that's not easy to achieve.

Relation to Prior Work: The paper is well positioned with respect to previous literature.

Reproducibility: Yes

Additional Feedback:


Review 4

Summary and Contributions: This paper describes a method for trajectory prediction of a target agent from a grid based birds-eye-view scene representation containing the static environment and surrounding dynamic objects. Key is that the method learns a shared latent space, with mappings from both scene representations, and from future agent trajectories. After mapping an input scene to this space, potential futures from a large trajectory bank can be matched efficiently using approximate nearest neighbor search. This approach is cast in a probabilistic kernel-density like formulation, such that a proper distribution over possible future is obtained for various inference tasks (sampling, mode/mean estimation). An extension for multi-modal distributions is also presented. Experiments are shown on the public Argoverse and an very large in-house data set (250K and 2M trajectories in the bank respectively). An ablation study on the in-house data set underline the assumptions, and the method outperforms previous in-house solutions (no state-of-the-art methods are cited here). On the Argoverse benchmark the method is in the top-4.

Strengths: * The idea of learning a shared mapping between the scene representation and future track representations seems novel for this problem setting, and is more akin to image/document retrieval-based methods than to existing methods in this field. * The rigorous commitment to ensure proper probabilistic inference is also significant, and an important aspect that many other methods still ignore but this paper rightfully underlines, also in the experiments and metrics. * Experiments are done on both the public Argoverse data set, as well as on a much larger proprietary in-house data set * The problem of multi-modal trajectory prediction is relevant for autonomous driving. This work also explicitly addresses the efficiency of inference, and important aspect for any practical application.

Weaknesses: The method is in the top-5 out of 61 reported results on the public Argoverse benchmark, which is probably good, but a bit more context should be given, possibly in the text. In what range is the top-10 ? In line 301 it is argued that VectorNet uses a more sophisticated scene representation, but which of the compared methods are then fair baselines with a similar scene representation? * No or limited insights in the properties of the in-house data set are given, and the baselines on this data set are created in-house too. The impact of the in-house benchmark would have been much stronger of one or a few state-of-the-art methods would have been tested too. * Inherently, this is a non-parametric density representation that requires all training data. In general, it is not surprising that a nearest neighbor approach outperforms any parametric model with sufficiently large data sets (in the limit of infinite training data, the nearest neighbor solution should be optimal), but in many practical applications storing all training samples for test time is impractical. I would have liked to see some attention to this trade-off, e.g. how does the amount of training data and/or trajectory bank affect the performance? Do you need such large data sets, is there still much to gain? What are the storage requirement for runtime inference?

Correctness: The overall approach seems sound, and also well-founded mathematically. A few comments: 1. At the start of section 3.2, the argument is made for a uniform h(t) since that makes the mode of the posterior P(t|q) more easily computable. While the mathematical argument is sound, it is not self-evident that a uniform h(t) actually yields a _useful_ P(t|q): should we want a uniform "prior" h(t) ? Wouldn't we in practice want the prior to emphasize the common trajectory patterns in the training set? The motivation could receive some more attention. 2. There is a lot of randomness used in the presented approach. How robust is the method to different random seeds for the k-means clustering and Monte Carlo sampling? Have results been repeated for multiple seeds? Can you report variance in your results? 3. Unless the in-house data set is released, many results cannot be reproduced.

Clarity: Generally the paper is well written, but what wasn't clear to me until the experiments section is that this scene description q contains both the static and the (history) of the dynamic objects, and thus implicitly captures the past trajectory of the current agent. When reading the Approach in section 3, I interpreted the scene description as a static description of the environment (i.e. static objects, maybe current position of other agents), and it wasn't clear that the description would include dynamic information too. I therefore didn't understand why the prediction wouldn't at least also be conditioned on the agent's past track in a supposedly similar trajectory space. In any case, this point can easily be clarified in section 3. An intuitive illustration of the method early on (like Figure 1) would help too. * The end of Equation (2) includes an $\rightarrow maxa,f,g$ which I find confusing as it isn't part of the r.h.s. of the log likelihood that the equation defines. I guess is intended to show that the objective is to maximize the log likelihood, but it doesn't make sense in the context of the running sentence. * Lines 138 and 142 first introduce the concept of uniformity of h(t), but it would be helpful to clarify the intended meaning of 'uniform' in this context a bit better: uniform over what? As I understand, you do not mean that h(t) should be a uniform distribution over the tracks in the training set (that would be trivial), but you want h(t) be uniform over the track space T but exclude regions of the track space that are unlikely or impossible, right? * line 187: I don't understand this sentence. Aren't all trajectories per definition from T = RMx2 (see line 90) ?

Relation to Prior Work: Yes

Reproducibility: Yes

Additional Feedback: EDIT: after reading the rebuttal and reviewer discussion, I am still overall positive about this work, and I congratulate the authors with their updated results, and their top-3 ranking with the ADE-1/FDE-1 scores on the Argoverse leaderboard (at the time of writing this edit). Still, one of my main concerns was the impact on the size of the training data, and upon re-evaluation of this work this remains a big concern. The rebuttal indicates that the method does not work well on the smaller pedestrian datasets. Additionally, upon rechecking the Argoverse benchmark, the method does not perform well in terms of Miss-Rate (MR) which is the main metric used to sort the leaderboard, since according to the Argoverse website "[MR] measures the number of scenarios where none of the forecasts are within 2 meters of the ground truth according to endpoint error. To make progress on this metric, entries need to focus on the most challenging scenarios, rather than minimizing errors on “easy” scenarios.". Exploring the limitations of the proposed approach is therefore crucial to better assess the value of the approach. I am lowering my review rating accordingly. --- * All tracks in the are assumed to have the same duration M. What happens with tracklets or otherwise incomplete tracks with missing observations? Is this a problem in practice? If we exclude broken tracks, might we not miss important behaviors e.g. of partially occluded objects in busy scenes? * line 151, I think Euclidean should be capitalized, as a proper name * Please add citations where possible to the related in Table 3 * I would have liked to see some qualitative samples of the In-house data. For instance, I would expect that multi-modality would be more important than it appears in Table 1, but this may also depend on the situation in your in-house data set, e.g. are there many junctions where distinct multi-modal futures can occur, or are there mostly scenes with a single destination? * line 247-249: Can you also say something about the amount of training time that is needed for both data sets?

[Author Response · NeurIPS 2020]

Firstly, we would like to thank the reviewers for spending their time on reviewing this submission and for the valuable feedback that came from it. We'll do our best to address the pointed out issues in the revised version of the paper.

**Argoverse**   We'd like to address the concerns raised about the quality of the proposed method on the Argoverse benchmark. Shortly after submitting the paper we've discovered a methodological bug in our experiments: it turned out we were using a trajectory bank built from the in-house dataset in the evaluation of our method on the Argoverse dataset. We've fixed the mistake and re-evaluated the proposed method. The updated results can be seen in Table 1. We've also updated metrics for all other methods where a better result has been published on the Argoverse leaderboard [1] at the time we've updated our result (June 16th, 2020). In terms of top-1 prediction metrics there still are two entries [3, 4] slightly ahead after the update. As with VectorNet, a direct comparison with these methods is hard, as we use a more sophisticated decoding scheme, while they propose improved scene encoders. One entry, which, just as our method, uses a rasterized representation of the scene is «uulm-mrm» [2, 5], which we surpass in quality. We'd be happy to add missing details to the paper, as well as the results of an involuntarily study of whether the trajectory bank from one dataset can be reused on another.

**Better baselines**   It has been argued that the baselines we compared against on the in-house dataset can be stronger. We however argue that the comparison is fair as we propose a new decoding scheme that can work on top of any scene encoder, and current SotA methods either use a simple fully-connected decoder [4, 5] or a RNN, which seems to work better [7]. This should be clarified in the paper.

**Pedestrian trajectory prediction**   It has been suggested that the generality of the proposed method can be demonstrated by applying it to the pedestrian motion prediction problem as represented by datasets such as SDD, ETH or UCY. However the proposed method requires a large training set to build the trajectory bank from, and all these datasets are rather small. To the best of our knowledge, no large public dataset for the pedestrian motion prediction problem has been released as of yet, as highlighted by works such as [6]. We have, however, tried our method on a large scale in-house dataset for pedestrian motion prediction and the conclusions were similar to what we've got for vehicles, so we decided not to add it to the paper as it doesn't add any new insights. We can add these results if deemed necessary.

**A study of the effects of clustering**   It has also been suggested that we should do more experiments to clarify the role the clustering procedure has on the induced $h(t)$ and the quality of the proposed method. We have experimented with various clustering schemes and hyperparameters and found no significant effect on the results, as long as there is some clustering. Some results obtained for a model that is slightly different to the one in the paper can be seen in Table 2. We'd argue that the specifics of the clustering procedure do not have a large effect on quality because the actual values of $h(t)$ become less relevant if we consider large enough number of max inner product search results, as we'll get most of the significant terms of the posterior anyway. We agree that this should be clarified in the paper, and a study of the effects of clustering should be added.

# References

[1] https://evalai.cloudcv.org/web/challenges/challenge-page/454/leaderboard/1279
[2] ArgoAI challenge results https://slideslive.com/38923162/argoai-challenge
[3] Mercat et al. "Multi-Head Attention for Multi-Modal Joint Vehicle Motion Forecasting." arXiv. 2019.
[4] Liang et al. "Learning Lane Graph Representations for Motion Forecasting." ECCV 2020.
[5] Cui et al. "Multimodal Trajectory Predictions for Autonomous Driving using Deep CNN." ICRA 2019.
[6] Jain et al. "Discrete Residual Flow for Probabilistic Pedestrian Behavior Prediction." CoRL 2019.
[7] Hong et al. "Rules of the Road: Predicting Driving Behavior with a Convolutional Model of Semantic Interactions." CVPR 2019

Table 1: Updated results on the Argoverse dataset

|  | Model | ADE@1 | FDE@1 |
|---|---|---|---|
| ADE@1 leaderboard top | Jean [3] | 1.68 | 3.73 |
|  | _anonymous (LGN) [4] | 1.71 | 3.78 |
|  | PRANK (ours) | 1.73 | 3.82 |
|  | poly | 1.77 | 3.95 |
|  | UAR | 1.86 | 4.09 |
| other | VectorNet | 1.81 | 4.01 |
|  | PRANK (ours, old) | 1.84 | 4.05 |
|  | uulm-mrm [2, 5] | 1.90 | 4.19 |

Table 2: A study of the effects of clustering

| Clustering configuration | N clusters | ADE | FDE |
|---|---|---|---|
| hierarchical compl.-linkage | 1k | 1.322 | 3.028 |
|  | 5k | 1.314 | **3.007** |
|  | 10k | 1.315 | 3.030 |
| k-means, implementation 1 | 10k | **1.313** | 3.011 |
| k-means, implementation 2 | 10k | 1.321 | 3.022 |
| k-means, implementation 3 | 10k | 1.322 | 3.031 |

[Meta-Review · NeurIPS 2020]

This paper generated extensive discussion with strong arguments in favor and against. Arguments against focused on minimal comparisons to existing literature and thoroughness of evaluation. However, many of the concerns are addressed in the rebuttal, and this does appear to be an interesting and novel solution to a hard problem.